

# Synergistic effects of climate and landscape change on the conservation of Amazonian lizards

Cássia de Carvalho Teixeira[1], Leonardo Carreira Trevelin[1,2], Maria Cristina dos Santos-Costa[3], Ana Prudente[1] and Daniel Paiva Silva[4]

[1] Programa de Pós Graduação em Biodiversidade e Evolução, Museu Paraense Emílio Goeldi, Belém, Pará, Brazil
[2] Instituto Tecnológico Vale - Desenvolvimento Sustentável, Belém, Pará, Brazil
[3] Laboratório de Ecologia e Zoologia de Vertebrados, Instituto de Ciências Biológicas, Universidade Federal do Pará, Belém, Pará, Brazil
[4] COBIMA Lab, Departamento de Biologia, Instituto Federal Goiano - Campus Urutaí, Urutaí, Goiás, Brazil

Corresponding author
Cássia de Carvalho Teixeira,
cassiacteixeira@gmail.com

## ABSTRACT

The leading causes of the worldwide decline in biodiversity are global warming, allied with natural habitat loss and fragmentation. Here, we propose an analysis of the synergistic effects of these two factors in 63 species of Amazonian lizards.
We predicted that the high-climatic suitability areas of species would be significantly impacted by different deforestation scenarios and the resultant landscape structure and considered that forest-dwelling species would be especially susceptible to deforestation scenarios. We also pointed out species threatened by both drivers and suggested critical areas for their future conservation. According to our results, most species will face future reductions in suitable areas for their occurrence according to five different patterns, two of which represent significant risks for 15 species.
Some of these species already deal with severe habitat loss and fragmentation of their current distribution ranges, whereas others will suffer a considerable area reduction related to future range shifts. We emphasize the importance of protected areas (PAs), especially indigenous lands, and the need to plan combined strategies involving PAs' maintenance and possible implementation of ecological corridors. Finally, we highlight eight species of thermoconformer lizards that constitute present and future conservation concerns related to the combined effects of climate change and habitat loss and that should be carefully evaluated in extinction risk assessments.

## INTRODUCTION

Anthropogenic global warming and climate changes, allied with the loss and fragmentation of natural habitats, are the main processes responsible for the worldwide decline in biodiversity (*Parmesan, 2006*; *Gallant et al., 2007*). Although often approached independently, their combined effects may produce synergistic outcomes that need to be accounted for to accurately evaluate the anthropogenic footprint on biodiversity

(*Mantyka-Pringle, Martin & Rhodes, 2012*; *Frishkoff et al., 2016*). For instance, cumulative evidence suggests a climatic resilience threshold for the Amazon Forest, where deforestation may hinder most tree species from geographically tracking the future climate and driving them to extinction, where climatic conditions turn unsuitable (*Cowling et al., 2004*; *Feeley & Silman, 2016*; *Gomes et al., 2019*). Climate-driven latitudinal range shifts (*Wilson et al., 2005*; *Hickling et al., 2006*) and potential extinction due to the failure to track suitable climate conditions (*Sekercioglu et al., 2002*; *Pounds et al., 2006*) may also be expected for animal taxa.

The historical concomitant increase in atmospheric concentrations of $CO_2$ and global surface temperature is projected to continue rising throughout the 21st century, irrespective of the greenhouse gas emission scenario considered (*IPCC, 2014*). The resulting changes in regional precipitation patterns have been observed in the continental-scale Amazon, showing an intensification of its hydrological cycle in the last decades, with more frequent seasonal extremes like floods and droughts (*Gloor et al., 2013*; *Chaudhari et al., 2019*). The Amazon forests naturally exert control over rainfall and temperature through evapotranspiration, especially attenuating drier climatic extremes (*Sampaio et al., 2018*). Still, the synergic interactions of the consequences of these climate projections with landscape change events, such as deforestation and fire events, may limit this mechanism and result in adverse effects on biodiversity (*Marengo et al., 2018*). Local-scale deforestation is predicted to change local climatic conditions, such as mean temperature or rainfall frequency (*D'Almeida et al., 2007*). Regionally, a threshold in deforestation, beyond which forests would no longer sustain their climate and collapse into drier savanna-like physiognomies is discussed (*Marengo et al., 2018*; *Lovejoy & Nobre, 2018*).

When considering deforestation, it is important to acknowledge that habitat loss and fragmentation have long been understood as two different landscape processes with varying effects on biodiversity (*Fahrig, 2003*). While less habitat area increases the likelihood of stochastic extinction and declines in population sizes, fragmentation refers to dividing the remaining habitat and isolating populations with limited dispersal capacity (*Fahrig, 1997*; *Fischer & Lindenmayer, 2007*). Despite this marked difference, both processes usually co-occur and interact non-linearly in landscapes (*Swift & Hannon, 2010*).

When habitat areas are abundantly available (*i.e.*, large forested areas), connectivity is still robust, and fragmentation effects are negligible. The importance of such effects only increases when the remaining habitat patches are so few and dispersed that isolation intensify (*Fahrig, 2002*). This situation was formalized as the extinction threshold hypothesis, which states that species require a minimum habitat area to persist in a landscape (*Andrén, 1994*; *Fahrig, 2013*). True fragmentation effects occur below this threshold, leading to abrupt declines in species populations and an increased chance of extinction (*Fahrig, 2002*; *Swift & Hannon, 2010*). Based on simulation studies, less than 30% of the original habitat area in the landscape was considered a critical threshold (*Andrén, 1994*). While it is demonstrated in empirical results that this value can vary among species, landscape, and spatial scale, this theoretical value is an important general

reference point for studies with scant empirical data (*Pardini et al., 2010*; *Swift & Hannon, 2010*).

Globally, reptiles have been proposed to be the primary animal taxa negatively affected by habitat loss and fragmentation as temperatures rise (*Mantyka-Pringle, Martin & Rhodes, 2012*). Lizards have an ectothermic metabolism and control body temperature through intricate behavioural and physiological processes that directly impact locomotory activities, such as foraging and dispersal (*Camacho & Rush, 2017*). These features make them particularly vulnerable to the synergic consequences of climate and landscape change in tropical regions (*Sinervo et al., 2010*). Some species need direct sunlight to thermoregulate and are usually associated with open areas (thermoregulators). In contrast, other species related to closed-canopy formations do not have this requirement (thermoconformers), some of which are commonly found in the evergreen forest of the Brazilian Amazon (*Vitt et al., 2008*). Even in these more climatically stable tropical regions, thermoconformers are active at low body temperatures, not tolerating severe warming and thus more vulnerable to climate and landscape change (*Huey et al., 2009*).

In recent studies, the effects of climate change on the distribution and thermal physiology of some tropical species have been explored (*Diele-Viegas et al., 2018*; *Pontes-da-Silva et al., 2018*). Likewise, landscape-change effects on tropical lizard species have been approached in several studies, suggesting strong species-area relationships regulating these systems (*Silva, Santos-Filho & Canale, 2014*; *Palmeirim, Vieira & Peres, 2017a*). Still, no studies have correlated climate and landscape change and evaluated their effects on the distribution of lizard species. A common approach to assessing the impact of climate on biodiversity is to use species distribution models (SDMs). In SDMs, individual occurrences of the species are correlated with local climatic characteristics to predict other high-suitability areas and project different climatic scenarios for their occurrence (*Guisan & Zimmermann, 2000*; *Peterson, 2003*). Also, to these SDMs, additional layers of information can be added later to refine the results and better inform scenarios, including deforestation models (*De Marco et al., 2018*; *Santos et al., 2020*; *Miranda et al., 2021*).

We aimed to quantify the combined effects of climate and landscape change on the species in Amazonian tropical rainforests (*Avila-Pires, 1995*; *Ribeiro-Júnior & Amaral, 2016a*) to fill this knowledge gap and guide the effective conservation of Amazonian lizard species in Brazil. Our approach involved understanding future scenarios of climate suitability for Brazilian Amazonian lizards and how these may be influenced by deforestation and extinction thresholds. We hypothesize that, given evidence of temperature-reduced locomotory performance (*Diele-Viegas et al., 2018*) and area effect-related local extinctions (*Palmeirim, Vieira & Peres, 2017a*), severe landscape changes may hinder Amazonian lizard species' dispersal and limit their abilities to track suitable conditions, leading to greater extinction risk (*Urban, 2015*; *Frishkoff et al., 2016*).

We predict that the potential climatic niche of the species will be significantly impacted by different deforestation scenarios and the resultant landscape structure. We further discuss how forest-dwelling species will be especially susceptible to such deforestation scenarios. Ultimately, we pointed out species threatened by the synergy between climate

and landscape change, drawing geographic and temporal patterns of species responses and prioritising conservation measures specifically driven by these patterns.

## MATERIALS AND METHODS

### Species occurrence dataset

We initially defined the Amazon basin as the extent of our study region (*Soares-Filho et al., 2006*; Fig. 1). Next, we created a list of lizard species endemic to this region and whose distributions are predominantly Amazonian within Brazilian borders (>50% of the distribution). Species distributions and the compilation of their occurrence records were based on the comprehensive works of *Ribeiro-Júnior & Amaral (2016a)* and *Ribeiro-Júnior (2015)*, which represent extensive taxonomical efforts of specimens deposited in scientific collections. When not directly available, geographical coordinates within these works were accessed with the aid of gazetteers and geolocation software based on historical information and collector's field notes, some of which correspond to municipality centroids. Finally, we assumed a minimum of 10 valid occurrences from this initial set as our primary criterion for modelling. The resulting species pool contained 63 Amazonian lizards, 50 of which were endemic to the Amazon (Table S1). The majority of species (53) had up to 35% of occurrences corresponding to municipality centroids, whereas for the remaining 10 species, up to 59% were municipality centroids. Despite the lower resolution of these specific occurrence data, we opted to keep them, as exclusion would cause a considerable loss of historical distributional knowledge of these species. An essential caveat within our approach is that occurrence data compiled from scientific collections contain a time lag between the time lizards were collected and present-day land-cover conditions. Still, we feel this is precisely a good reason to justify using additional present landcover data as a means to refine the suitability outputs from the initial SDMs we analyzed.

### Modeling climate scenarios

We performed species distribution modelling using the species occurrence dataset and environmental variables representing two climatic scenarios to estimate the impact of climate change on the geographic distribution of Amazonian lizard species: current and future (Table 1).

Considering the current scenario within our predefined extent for analysis, we used 19 bioclimatic variables from WorldClim (*Hijmans et al., 2005*; http:\\www.worldclim.org) under a 20.25 km$^2$ (~4 km side = 2.5 arc-min) grid resolution. Highly correlated variables were arbitrarily removed before modelling, using Person's correlation ($r > 0.8$) to avoid collinearity in our models (*Guisan & Thuiller, 2005*) (Table S2). The selected variables were: Annual Mean Temperature (BIO1), Mean Diurnal Range (BIO2), Isothermality (BIO3), Temperature Seasonality (BIO4), Annual Precipitation (BIO12), Precipitation of Driest Month (BIO14), Precipitation of Warmest Quarter (BIO18) and Precipitation of Coldest Quarter (BIO19).

When considering the future scenario, we used the same bioclimatic variables corresponding to the year 2050, this time derived from 17 Atmosphere-Ocean coupled

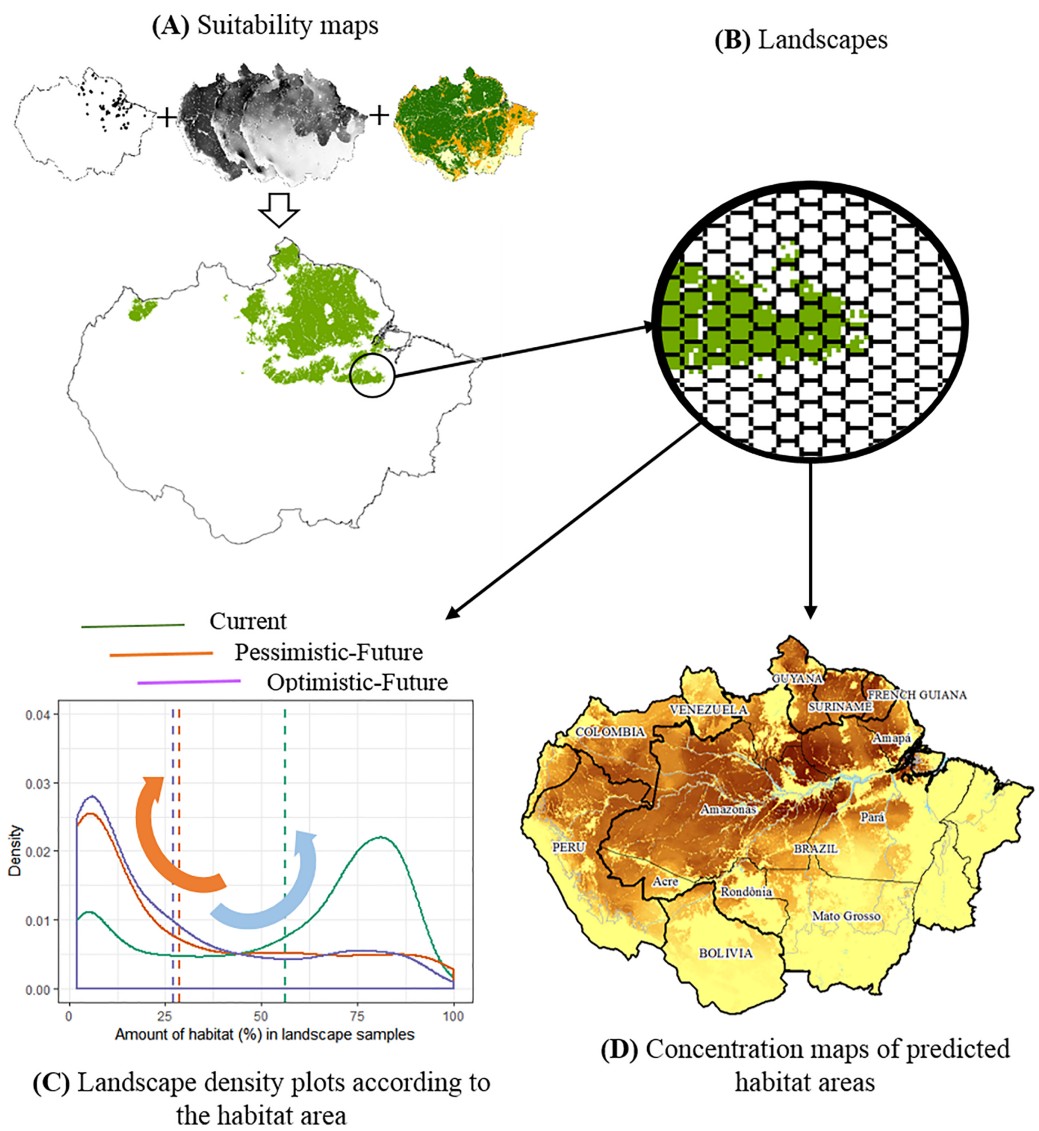

**(A)** Suitability maps

**(B)** Landscapes

Current
Pessimistic-Future
Optimistic-Future

**(C)** Landscape density plots according to the habitat area

**(D)** Concentration maps of predicted habitat areas

**Figure 1** **The conceptual model used to evaluate the joint effects of habitat loss and climate change in the potential distribution of Amazonian lizards.** (A) Modeling suitability surfaces and defining potential distribution. (B) Division into landscape samples and habitat area quantification. (C) Density plots to evaluate changes in the frequency distribution of landscape covers within the predicted distribution for each climatic/deforestation scenario. The red arrow represents the high concentration of landscapes with about 30% of habitat area cover below the critical threshold, while the blue arrow represents the high concentration of landscapes with >75% of habitat area cover. (D) Concentration maps of predicted habitat areas.

General Circulation Models (AOGCMs) also available from WorldClim: ACCESS1-0, BCC-CSM1-1, CCSM4, CNRM-CM5, GFDL-CM3, GISS-E2-R, HadGEM2-AO, HadGEM2-CC, HadGEM2-ES, INMCM4, IPSL-CM5A-LR, MIROC-ESM-CHEM, MIROC5, MPI-ESM-LR, MRI-CGCM3 e NorESM1-M. For all, we chose the most pessimistic carbon dioxide emission perspective (Representative Carbon Pathway—RCP 8.5 - *IPCC, 2014*). The pessimistic perspective was used, as it has recently been assumed to strongly agree with the historical cumulative $CO_2$ emissions (*Schwalm, Glendon & Duffy,*

**Table 1 Climate + forest cover scenarios.** Scenarios combining effects of climate and landscape change and how they were defined in this study, and the observed response patterns depicting how habitat availability in the landscape changes in each scenario, for each pattern.

| Moment | Climate change scenarios | Deforestation scenarios | Observed response patterns | | | | |
|---|---|---|---|---|---|---|---|
| | | | Pattern 1 | Pattern 2 | Pattern 3 | Pattern 4 | Pattern 5 |
| Current (2019) | SDMs based on 8 uncorrelated bioclimatic variables derived from the monthly temperature and rainfall values describing local, current climatic trends (BIO01, BIO02, BIO03, BIO04, BIO12, BIO14, BIO18, BIO19[*]) | Forest cover projections on a governance scenario (GOV - 2019): environmental governance scenario projecting deforestation similar to the actual rates of the present days[**] | Higher density of landscapes with >75% habitat area | Similar density of landscapes with >75% and <30% habitat area | Higher density of landscapes with <30% habitat area | High density of landscapes with >75% habitat area | High density of landscapes with >75% habitat area |
| Future (2050) | Consensual SDMs based on the same 8 bioclimatic variables, derived from 17 AOGCMs, considering the most pessimistic $CO_2$ emission perspective (RCP 8.5). (BIO01, BIO02, BIO03, BIO04, BIO12, BIO14, BIO18, BIO19[*]) | Forest cover projections on a governance scenario (GOV - 2050): Optimistic, environmental governance scenario projecting 17% forest loss in the year 2050[**] | Higher density of landscapes with >75% habitat area | Similar density of landscapes with >75% and <30% habitat area | Higher density of landscapes with <30% habitat area | Similar density of landscapes with >75% and <30% habitat area | Higher density of landscapes with <30% habitat area |
| | | Forest cover projections on a "business-as-usual" scenario (BAU - 2050): Pessimistic, low environmental compliance scenario, projecting 37% forest loss in the year 2050[**] | Higher density of landscapes with >75% habitat area | Similar density of landscapes with >75% and <30% habitat area | Higher density of landscapes with <30% habitat area | Similar density of landscapes with >75% and <30% habitat area | Higher density of landscapes with <30% habitat area |

Notes:
[*] Bioclimatic variables extracted from WorldClim (*Hijmans et al., 2005*; https://www.worldclim.org/).
[**] Forest cover rasters extracted from SimAmazonia project (*Soares-Filho et al., 2006*; https://csr.ufmg.br/simamazonia).

*2020*). Lastly, we used an arithmetic mean of the resulting predictions as the consensual future model for each species, a robust approach to this end (*Marmion et al., 2009*).

The original occurrence datasets contained presence-only data. Therefore, we generated pseudo-absence data randomly distributed for each species (*Lobo, Jiménez-Valverde & Hortal, 2010*). Then, we used the "checkerboard" methodology to partition data approximately in half between two spatially structured subsets, divided into cells resembling a checkerboard table. Each subset was used to both fit the model and validate the model fitted onto the other subset, a common approach in evaluating species distribution models (*Muscarella et al., 2014*; *Velazco et al., 2017*).

We generated SDMs using three machine learning algorithms, regarded as highly efficient for modelling species distribution (*Elith et al., 2006*; *Meynard & Quinn, 2007*; *Oppel et al., 2012*): the "Maximum Entropy (MaxEnt)", based on the maximum entropy principle (*Phillips, Anderson & Schapire, 2006*); the "Random Forest", which separates data into randomly defined decision trees and elect the most recurrent and likely distribution (*Schapire, 2001*); and the "Support Vector Machine (SVM)" method based on probability classes (*Guo, Kelly & Graham, 2005*; *Phillips, Anderson & Schapire, 2006*). Adopted

models' restrictions resulted in few species with restricted distributions (<25) within our species pool. Thus, we did not feel the need to minimize the chance of omission error (in detriment to the commission error). ROC generated slightly less inflated SDMs, allowing us to perceive the reported patterns better. We used thresholds obtained with the "Receiver—Operator Curve" (ROC), which balances both omission and commission errors while determining distributional ranges (*Phillips, Anderson & Schapire, 2006*), to transform climatic suitability predictions into presence/absence distribution maps.

We used two approaches to evaluate the performance of the generated models: the "Area Under the Receiver Operator Curve" (AUC) which varies from 0 to 1, with values ≤0.5 representing models no better than random; and the "True Skilled Statistics" (TSS), which varies from −1 to 1, with values close to or below 0 also representing models no better than random. We considered models that scored above 0.7 in both statistics as acceptable (*Allouche, Tsoar & Kadmon, 2006*; *Liu, White & Newell, 2011*). This modeling approach was implemented in the R 3.6.0 environment (*R Core Team, 2017*), using the ENMTML package (*Andrade, Velazco & De Marco Júnior, 2020*). Ultimately, we identified priority conservation areas coinciding with the highest amount of the analysed species.

## Forest cover scenarios

We refined current and future climate change scenarios by overlapping forest cover rasters generated with a deforestation model developed by the SimAmazonia project (https://csr.ufmg.br/simamazonia/; *Soares-Filho et al., 2006*) to further improve the SDMs of each species and evaluate the synergic effects of climate and landscape change. Based on data from the "Measurement of Deforestation by Remote Sensing in Amazonia Program - (PRODES)" (http://terrabrasilis.dpi.inpe.br/en/home-page/), cartographic algebra and cellular automata techniques are used by this model to simulate spatially-explicit deforestation models across the Amazon basin within two contrasting management scenarios: the "business-as-usual" (BAU) and the "governance" (GOV) scenarios (Table 1). BAU is a pessimistic scenario that assumes low compliance with environmental laws and the paving of major highways, projecting rates of deforestation based on historical figures that result in an expected 37% forest reduction up to 2050. On the other hand, GOV is an optimistic, environmental governance best-case scenario that projects deforestation rates as an asymptotic logistic curve, imposing limits on the deforested land that aims to reproduce public policies, resulting in estimates of approximately 17% of deforestation for the same year 2050 (*Soares-Filho et al., 2006*).

We used both SimAmazonia projections to refine future climate scenarios and a single SimAmazonia projection to refine the current present-day scenario (2019) instead of the actual data. This is because the actual forest cover data are only partially available for some parts of the study region. By doing this, we guaranteed the full extension of the analyzed area and maintained standardisation for the temporal comparisons. We visually identified that the GOV projections for 2019 were closely related to actual forest cover data and used them as our current scenario (Table 1).

## Habitat area in the landscape

We used nearest neighbour resampling to downscale all current and future forest cover projections, initially available in one km$^2$ resolution, to the same resolution matching the bioclimatic SDMs. Then, we overlapped SDMs and forest cover projections and calculated the total number of high-climatic suitability cells coinciding with predicted forested cells, assuming them as potential habitat areas, which were estimated for each species in each scenario. Habitat area is hypothesized as a critical predictor of biodiversity in landscapes (*Fahrig, 2013*) and here we also evaluate habitat distribution in the landscapes. Thus, each resulting habitat area prediction was gridded in a tessellation of hexagonal cells, with 1,200 km$^2$ each, representing a sample of a landscape for the studied lizards for which we counted the number of available habitat cells. Samples with these dimensions have been previously used to represent local landscapes for amphibian species (*Becker et al., 2010*). Given the known spatial and energetic requirements of lizards (*Pough, 1980*), we considered it a reasonable estimate of the habitat area in the landscape for the studied lizards. According to the extinction threshold hypothesis (*Andrén, 1994*), we considered those landscapes with at least 30% habitat as stable landscapes. As a result, we generated estimates of total habitat areas and total stable habitat areas (only considering those landscapes >30% of forested habitats) for each species by each scenario.

Furthermore, we assessed the vulnerability of the studied species by evaluating the total stable habitat areas within the Brazilian System of Protected Areas (PAs). For this, we adopted the criteria proposed by *Rodrigues et al. (2004)*: species with <1,000 km$^2$ are expected to be 100% within PAs; species with >250,000 km$^2$ are expected to be at least 10% within PAs. In between, the percentage is calculated based on logarithmic interpolation. Species that did not meet these criteria were considered vulnerable. It is important to remember that 11 out of the 63 studied species are not endemic to the Amazonia biome, so our approach assessed the Amazonian status of these species.

We considered all three categories of PAs in our analysis: (a) full protection areas, whose sole objective is to preserve biodiversity without any exploitation of natural resources, (b) sustainable use areas, destined to conservation and sustainable use of natural resources and (c) indigenous lands destined to the protection of indigenous populations, but with a relevant role in biodiversity protection complementing the other PAs (*Begotti & Peres, 2020*). We used the Protected Planet database as our source for this data (www. protectedplanet.net; *UNEP-WCMC, 2020*). Lastly, we summed the stable habitat areas for each species by scenario, aiming to identify the richest and climatically stable locations as priority areas for conserving Amazonian lizards.

## Data analysis

We added the SDMs (high-climatic suitability cells refined by total stable habitat areas) of all species to visualise priority areas for the conservation of lizard species across the Amazon basin. This procedure resulted in species richness predictions for each climate + forest cover scenario (Figs. 2A and 2B; Figs. S1A and S1B). We also used the same approach and produced richness predictions only for the most vulnerable species identified in our study (paragraphs below). Therefore, we indicate priority areas for

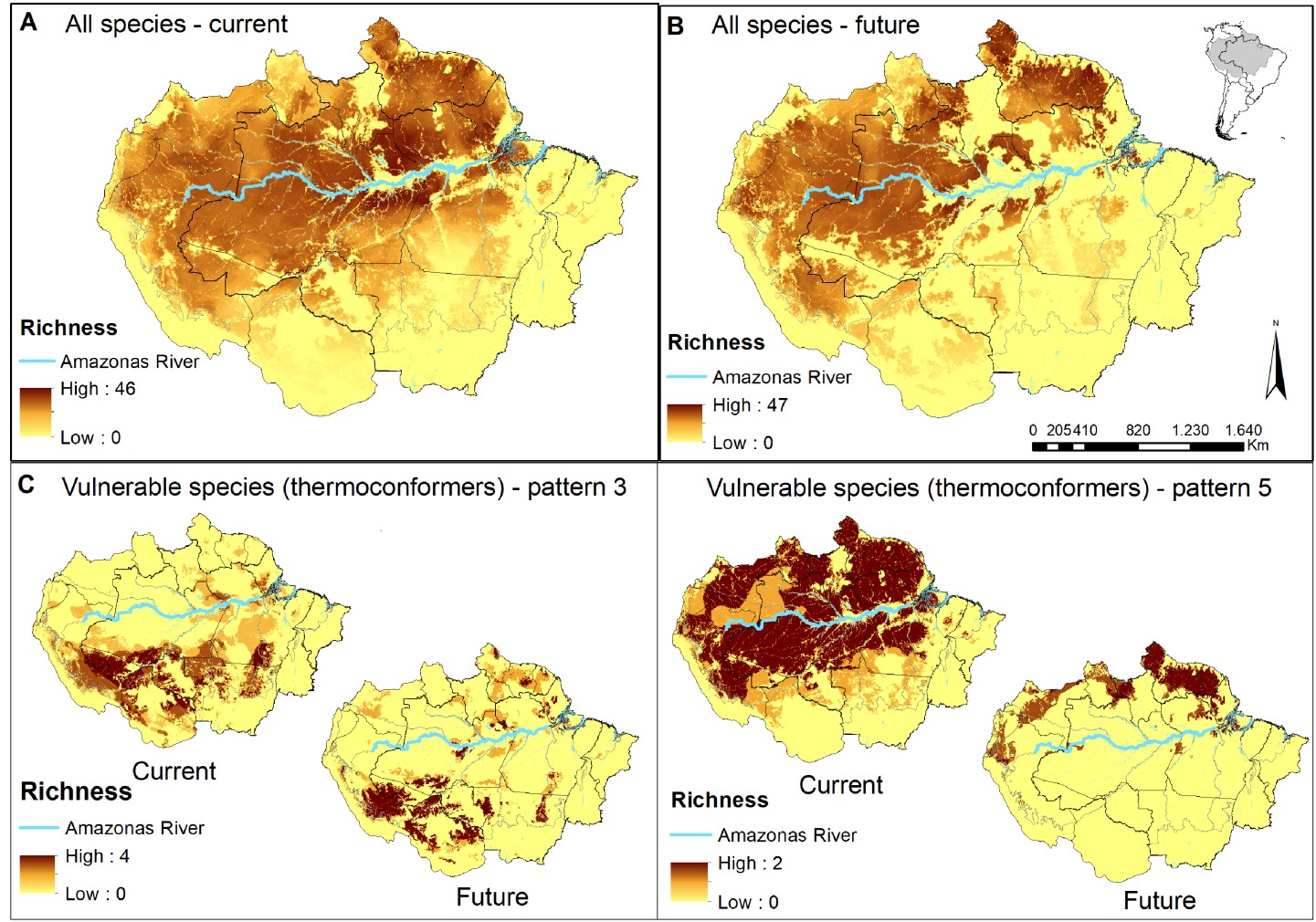

**Figure 2 Concentration maps of predicted habitat areas.** (A) Species richness based on the sum of habitat areas for all Brazilian species in the current and (B) in a future pessimistic scenario, depicted by a gradient of colours, from yellow (no overlap) to brown (maximum overlap of species). The same approach only considering thermoconformers' vulnerable species classified on (C) pattern 3 and (D) pattern 5. Blue highlight corresponds to the Amazon river.

conserving the most vulnerable species to the synergic effects of future climate and landscape changes (Figs. 2C and 2D; Figs. S1C and S1D).

Next, we used species as replicates in each studied scenario to evaluate the overall effect of deforestation and extinction thresholds on the predicted distribution of lizard species. To this end, we compared the mean difference between the total habitat area and the total stable habitat area (landscapes >30% habitat threshold), using a non-parametric paired Wilcoxon signed-rank test.

For each species, we created density plots to evaluate changes in the frequency distribution of habitat area in landscapes within the predicted distribution for each climatic + forest cover scenario. These plots aid in visualising the shape of data distributions over continuous intervals (*i.e.*, landscape habitat cover), using kernel smoothers to plot values and minimize noises. Thus, we produced three data distribution
curves for each species: one for the present day (2019) and two for the future (2050),
considering pessimistic and optimistic scenarios. The peaks in these distributions aid us in
visually identifying where landscape samples concentrate, along with the habitat cover
interval. In this way, landscapes with more than 75% of habitat area suggest larger,
continuous habitats, whereas landscapes with less than 30% of habitat area suggest smaller,
more fragmented habitats beyond the theoretical extinction thresholds adopted in this
study (*Andrén, 1994*; *Fahrig, 2013*). We visually compared the shape of these distributions
in all scenarios for each species individually, evaluating whether habitat availability in
current present-day landscapes changes according to the proposed future scenarios
(Table 1). By evaluating distribution curves simultaneously, we aimed to identify patterns
of change that were repeated among species. Finally, we assessed the significance of
changes in frequency distributions using the Anderson–Darling test, a statistic to compare
two independent sample distributions, sensitive towards differences in the tails of
distributions (*Engmann & Cousineau, 2011*). The statistical significance was assessed by
permutation tests (*Quinn & Keough, 2002*), building null models containing 1,000
sub-samples randomly drawn from the original dataset. All analytical steps were
implemented in the R 3.6.0 environment (*R Core Team, 2017*), using the Ksamples package
(*Scholz & Zhu, 2019*). The complete methodological approach is summarized in a
flowchart depicted in Fig. 1, complemented by descriptions in Table 1.

## RESULTS

Among the 63 species evaluated in this study, *Stenocercus fimbriatus Avila-Pires, 1995*
showed the least number of records ($N = 10$) and *Ameiva ameiva* (Linnaeus, 1758) showed
the largest number of occurrences ($N = 493$) (Table S1). All species are considered to be
*Least Concern* in IUCN's Red List of threatened species (2018), while the species
*Colobosaura modesta* (Reinhardt & Luetken, 1862) and *Norops brasiliensis* (Vanzolini &
Williams, 1970) are considered to be threatened species in the state of Pará (Brazil)
(*Albernaz & Avila-Pires, 2009*). All models adjusted reached excellent predictive capacity
when evaluating their AUC (range 0.99–1.0) and TSS (range 0.72–0.92) (Table S1).

When evaluating our species richness predictions, the current highest concentration of
predicted habitat areas was centralized in our study region. The highest and largest species
richness concentration is found in the Amazon river's lowlands, in the transition between
the northern portions of the Brazilian states of Amazonas and Pará, whereas smaller
concentrations of high species richness can be found northeast, towards the Guyana region
(Fig. 2A). In the pessimistic future scenario, these high-concentration areas move north-
eastward, encompassing a wide stripe following the Atlantic coast with hotspots in the
Guyanas region and mouth of the Amazonas River (Fig. 2B). Around half of all habitat
areas were within PAs in all scenarios for all species (Table S3). These concentrations
represent areas of climatic stability for most Amazonian lizard species and provide a
general overview of their conservation status.

When comparing the mean difference between the total habitat area and the total stable
habitat area (Table S3), habitat area availability was significantly reduced in all tested
scenarios, with large effect sizes (Current: $v = 2016$, $p < 0.001$; $r = 0.87$; Pessimistic-Future:

$v$ = 2016, $p < 0.001$; $r = 0.87$; Optimistic-Future: $v$ = 2016, $p < 0.001$; $r = 0.87$). For current present-day projections, total habitat area reduced on average 2% (counted forest cells) when considering habitat only on landscapes above the theoretical extinction habitat cover threshold. We found an average decline of 9% for the pessimistic and 8% for the optimistic scenarios for future projections.

When considering species independently, most species presented declines in the availability of habitat areas in stable landscapes over time (Table S3). Out of the 63 species, 35 presented an average decrease of 32% in habitat area when considering the optimistic future scenario, while this number rises to 51 species with an average reduction of 42% in habitat area if we consider the pessimistic future scenario. Meanwhile, the remaining 28 species showed an average increase of 73% in the optimistic future scenario, and the remaining 12 species in the pessimistic scenario increased an average of 88% habitat area.

## Response patterns to climate and landscape change

We observed five different patterns of change in the frequency distributions of the habitat area in landscapes between scenarios. These patterns correspond to five different ways species respond to the synergic effects of climate and landscape changes over time (Fig. 3; Fig. S2).

In the first three patterns, the relative stability between the present and future scenarios is described. For example, pattern one ($N = 34$ species) corresponds to those species with a high density of landscapes with >75% of the habitat area in all scenarios, suggesting a low vulnerability to the combined effects of climate and landcover changes over time. Pattern two ($N = 4$) corresponds to species that display two similar peaks in the density of landscape habitat coverage (around 30% and above 75% of habitat area) that remain relatively stable between the present and future scenarios, also suggesting species with low vulnerability. On the other hand, pattern three ($N = 10$) corresponds to species that currently show a high density of poorly covered landscapes (below 30% habitat area), which tend to slightly increase in future scenarios, depicting the most vulnerable species in this study.

Meanwhile, in the other two patterns, variable scenarios over time are described. For example, in pattern four ($N = 10$) species with a higher density of landscapes with >75% of habitat area slightly shifting toward poorly covered landscapes in future scenarios are shown. These species shift to a two-peak density pattern of landscape habitat coverage in both future scenarios (similar to pattern 2), which still retain a similar amount of well-covered landscapes (>75%) and thus were considered at low vulnerability. Finally, pattern five ($N = 5$) corresponded to species that currently display a high density of landscapes with >75% of habitat area cover but are predicted to severely shift to poorly covered landscapes (>30% of habitat area cover) in the future scenarios. Such a condition suggests that these species do not currently display any vulnerability but can do so in the future.

Among the 63 studied species, our approach points to 10 species considered as currently vulnerable (pattern three) and to another five species that have the potential to become vulnerable in future climate and deforestation scenarios (pattern five). Together, it is

| Graph | Pattern | Responses | Amount of species | Threat level |
|---|---|---|---|---|
|  | 1 | High density of landscapes >75% of habitat area | 34 | Low |
|  | 2 | Landscapes <30% and >75% in similar proportions | 4 | Low |
|  | 3 | High density of landscapes <30%. | 10 | High |
|  | 4 | Small reduction of landscapes >75% in future scenarios | 10 | Low |
|  | 5 | High reduction of landscapes >75% in future scenarios | 5 | High |

**Figure 3** **Infographic representing the five general patterns recognized in the distribution of Brazilian Amazonian lizard species.** Density plots depict the frequency of landscapes along an axis of habitat area (%) observed for that pattern, a short description of the observed pattern, the number of species recognized within this pattern, and the threat category assigned to this pattern. The plots represent the distribution of *Norops fuscoauratus* (D'Orbigny, 1837 in Duméril & Bibron, 1837) (pattern 1), *Neusticurus bicarinatus* (Linnaeus, 1758) (pattern 2), *Colobosaura modesta* (pattern 3), *Kentropyx striata* (Daudin, 1802) (pattern 4) and of *Copeoglossum nigropunctatum* (Spix, 1825) (pattern 5), used as examples of each general pattern.

suggested in our approach that 23% of the studied species need special attention to mitigate the adverse synergic effects of climate and landscape changes.

The frequency distributions of landscape habitat covers in future scenarios were significantly different from their present state for most species across all patterns (Table S4). However, there was only one exception for the pessimistic scenario and five exceptions for the optimistic scenario (8% of species) (Table S4). These noticeable differences among the scenarios, variable among five patterns for all 63 species, highlight the relevance of designing informed public policies that aim to reduce the specific effects of climate and landscape change over biodiversity.

## DISCUSSION

In our results, we show the joint effects of climate and landscape change on 63 Amazonian lizard species, highlighting potential, previously overlooked, adverse effects on 15 of these species. We found a large extent of suitable area for all species, and almost half of these extents comprised PAs. This proportion inside PAs is similar to the results found by *Ribeiro-Júnior & Amaral (2016a)*, and we considered them to meet the conservation goals proposed by *Rodrigues et al. (2004)*. However, when limited to landscapes above 30% of the habitat area, the expected range of these species reduced significantly. The 15 abovementioned species display present, future, or both scenarios that could represent conservation concerns.

We corroborated our initial prediction by showing that for most species (51), an average reduction of 42% of the habitat area is expected by 2050. These results are consistent with the overall deforestation expected for the region (*Soares-Filho et al., 2006*). Even in an eventual optimistic future scenario, 35 species will still face an average 32% reduction in their expected habitat area. This average reduction is sufficient for us to suggest a risk of extinction according to the IUCN A4 criterion (*Gomes et al., 2019*; *IUCN, 2020*). There is considerable specific variation around these average values (minimum of 2% to a maximum of 97%), and some of the remaining species will benefit from future increases in the suitable area. Nonetheless, resulting from our 30% habitat area criterion, a group of 10 species already deals with habitat loss at the landscape level, which will be further aggravated until 2050 (pattern three). The distribution ranges of another five species will primarily shift to deforested landscapes in the future (pattern five).

*Palmeirim, Vieira & Peres (2017a)* studied lizard assemblages in an insular Amazonian landscape and found strong species-area effects. They observed functionally impoverished assemblages within smaller isolated islands, favoring larger-bodied thermoregulator species tolerant to forest edges and open areas. Similar results were also found for lizard assemblages in fragmented continental landscapes (*Bell & Donnely, 2006*; *Almeida-Gomes & Rocha, 2014*), and metabolism-related microhabitat use is discussed as the primary explanation for species turnover along a forest degradation gradient in studies across the Neotropics (*Vitt, Zani & Esposito, 1999*; *Palmeirim, Vieira & Peres, 2017b*). It is feasible to expect these same processes in landscapes with a low habitat area in our analyses (below 30%; *Andrén, 1994*; *Fahrig, 2003*). From this perspective, eight out of the 15

concerning species from our study are considered truly thermoconformators and are directly implicated with local landscape or future range shift extinction risks.

In pattern three, we recognized 10 species that already face habitat loss at the landscape level, which will be aggravated in future scenarios. Six of them are thermoconformers: *Enyalius leechii* (Boulenger, 1885), *Polychrus liogaster* Boulenger, 1908, *Stenocercus roseiventris* Duméril & Bibron, 1837, *Uracentron a. azureum* (Linnaeus, 1758), *Cercosaura eigenmanni* (Griffin, 1917) and *Neusticurus rudis* Boulenger, 1900. Relative smaller patches of habitat areas are found northeastern to the Amazon River, while the majority is widely distributed south of the river, overlapping to some extent with the region known as the "arch of deforestation," which indeed accounts for their current threats (*Fearnside, 2005*). Future scenarios predicted for species in this group do not represent major latitudinal or longitudinal shifts in the distribution of habitat areas but rather a "rarefying effect" within the current distribution (Fig. 2C). The exception is the northeastern patch dislocating towards the Guiana shield region. Approximately half the PAs within the potential distribution of these species are in indigenous lands, current strongholds of biodiversity (*UNEP-WCMC, 2020*; *Begotti & Peres, 2020*), extremely relevant to the conservation of these lizard species. Based on these patterns, we suggest that the strategy for PAs that currently harbor these species should receive significant attention in any conservation planning towards these taxa.

As for pattern five, we recognized that five species would be threatened in future scenarios by experiencing climate range shifts to predicted highly deforested landscapes. Two of these species, *Dactyloa punctata* (Daudin, 1802) and *Bachia flavescens* (Bonnaterre, 1789), are classified as thermoconformers. These species are predicted to have extensive habitat areas across the biome that will suffer a severe reduction in future availability, promoting a range shift toward the equator, especially in the Guiana shield (Fig. 2D). Despite being considered a knowledge gap in lizard distribution, 62% of the herpetofauna species found in this region are expected to be exclusive to it (*Hoogmoed, 1979*; *Avila-Pires, Hoogmoed & Vitt, 2007*). With our results, we show it as a stable climatic refugee, distant from deforestation frontiers and exceptionally relevant for species and general lizard conservation in Amazonia. In a business-as-usual scenario in Brazil, future conservation of these species will eventually depend on policies of other countries (Guiana, Suriname, French Guiana). Therefore, present-day conservation strategies should focus on allocating resources to promote connectivity between PAs (*Araújo et al., 2004*; *Millar, Stephenson & Stephens, 2007*), primarily because of the temperature-reduced locomotory performance expected for them (*Diele-Viegas et al., 2018*). Ecological corridors are favored in a densely forested region such as the north of the Amazon River, and are the most logical plan. Nonetheless, secondary forests and shaded plantations are effective ecological corridors for leaf-litter species in anthropogenic landscapes (*Dixo & Metzger, 2009*), as they are for other organisms (*Faria et al., 2007*; *Pardini et al., 2009*) and should be considered in future scenarios.

Ecological niche models assume equilibrium between species and the environment, thus not considering the ability of the species to adapt to new environmental circumstances (*Araújo et al., 2004*). Given the metabolic needs and microhabitat use of the species,

thermoregulators could display different and even positive responses to these same climate-landscape processes. *Frishkoff et al. (2016)* demonstrated for bird species that affiliation to drier climates is associated with an ability to persist in and colonize anthropogenic landscapes, arguing for the same homogenizing potential described for lizard communities (*Palmeirim, Vieira & Peres, 2017a*, *2017b*). Four thermoregulator species suggested as vulnerable in our analytical approach also occur in the savanna vegetation of the adjacent Cerrado biome: *N. brasiliensis*, *Hoplocercus spinosus* Fitzinger, 1843, *C. modesta* and *Tropidurus oreadicus* Rodrigues, 1987 (*Costa & Bérnils, 2018*; *Silva & Bates, 2002*). Furthermore, the latter, as well as *Tupinambis teguixin* (Linnaeus, 1758) and *Kentropyx calcarata* Spix, 1825 are discussed as tolerant to some level of habitat disruption, and both *T. oreadicus* and *T. teguixin* as perianthropic (*Lima, Suárez & Higuchi, 2001*; *Ribeiro-Júnior, 2015*; *Ribeiro-Júnior & Amaral, 2016b*; *Palmeirim, Vieira & Peres, 2017b*). These species should also be monitored. We suggest further studies to clarify whether they can be used as indicators of climate and landscape changes, considering their peculiarities (*e.g.*, the known association of *T. teguixin* to water bodies; *Avila-Pires, 1995*).

Our findings represent an original contribution towards understanding the expected consequences to Amazonian reptile biodiversity (*Cordier et al., 2021*) that arise from the expected negative feedback between changes in the landscape and regional climate of Amazonia (*Marengo et al., 2018*). New studies that aim for the underlying ecological mechanisms resulting in the pessimistic outcomes we described for some species would certainly further explore these results and refine conservation actions (*Schulte to Bühne et al., 2021*). From the landscape change perspective, new approaches breaking apart the processes involved in the extinction threshold hypothesis and studying the independent effects of habitat area and connectivity at a landscape scale would better inform management actions for specific taxa (*Miranda et al., 2021*). From the climate change perspective, more recent climate predictions (Coupled Model Intercomparison Project Phase 6, CMIP6, – https://www.wcrp-climate.org/wgcm-cmip/wgcm-cmip6) would certainly refine and improve future scenarios (although only available for 2040 and 2060). Finally, improvements in lizard taxonomy may certainly reveal further obscured patterns, as species like *Bachia flavencens* e *Neusticurus rudis* are already considered as potential complexes of species (*Kizirian & McDiarmid, 1998*; *Kok et al., 2018*), which probably results in the division of expected distributions and new, more endangered species (*Bickford et al., 2007*).

## CONCLUSIONS

Despite the significant knowledge gaps regarding the Amazonian lizard species we analysed here, we demonstrated that using the current resolution in the knowledge on species occurrences and a simulation-based habitat threshold was an appropriate approximation to elucidate the variable synergic effects through which climate and landscape change can potentially affect them. Researchers should focus on specific species suggested here (*Enyalius leechii*, *Polychrus liogaster*, *Stenocercus roseiventris*, *Uracentron a. azureum*, *Cercosaura eigenmanni*, *Neusticurus rudis*, *Dactyloa punctata* and *Bachia*

*flavescens*) as primary vulnerable species, whose extinction thresholds may vary according to specific natural history. Nonetheless, empirical data generally corroborated the simulations in their ability to identify vulnerable taxa (*Swift & Hannon, 2010*; *Mendes & De Marco Júnior, 2018*). The eight thermoconformer lizards highlighted here represent present and future conservation concerns that should be carefully evaluated in extinction risk assessments. We propose mitigation strategies related to their specific response patterns to climate and landscape combined effects, complementing species representations in current PA networks and promoting connectivity between them (*Araújo et al., 2004*; *Hannah et al., 2007*; *Millar, Stephenson & Stephens, 2007*). Early response is the best cost-saving and cost-benefit strategy for anticipating climate change (*Hannah et al., 2007*).

Worldwide, water availability is changing significantly, and South America is expected to experience significant hydric stress in the medium term (*Rodell et al., 2018*). Additionally, the current greenhouse gas emissions levels indicate that the global system will follow an irreversible pathway of global climate change ("Hothouse Earth"). In this sense, the Amazon Forest has an essential role as a sink for carbon dioxide, whereas decreasing greenhouse gas emissions are of utmost importance and priority (*Steffen et al., 2018*). Still, the current political scene in Brazil—the country bearing the most significant portion of the Amazon Forest—indicates a pathway going in the opposite direction from sustainability. Anthropic activities, such as cattle raising, crop plantation, and mining activities, account for most of the deforestation rates in the Amazon basin (*Soares-Filho & Rajão, 2018*; *Dobrovolski et al., 2018*). The current rates observed this year (in the first quarter of 2021) were higher than those observed in the previous year and the most extensive rates observed in the last years (*IMAZON, 2021*). Therefore, the climatic and landscape changes expected to affect the Amazonian lizards we evaluated here could be underestimated. Additionally, we also believe that other environmental factors, such as the effects of fire events (*Marengo et al., 2018*), will undoubtedly contribute to harsher scenarios for these lizard species.

## ACKNOWLEDGEMENTS

We thank Leandro Juen, Levi Carina Terribile, and "The METALAND Lab" at Universidade Federal de Goiás (UFG), especially Arthur Bispo, André Andrade, and Paulo de Marco Jr., for their help at different stages of this study.

### Funding

Cássia Teixeira received a scholarship from the Conselho Nacional de Desenvolvimento Científico e Tecnológico (CNPq - 142457/2014-0) and financial support by the Programa Nacional de Cooperação Acadêmica (PROCAD 2013, n°071/2013). Daniel de Paiva Silva and Ana Prudente received a productivity grant from Conselho Nacional de Desenvolvimento Científico e Tecnológico (CNPq proc. number: 304494/2019-4 and 302611/2018-5, respectively). This work was developed in the context of the National

Institute for Science and Technology (INCT) in Ecology, Evolution, and Biodiversity Conservation, supported by MCTIC/CNPq (proc. Number 465610/2014-5) and FAPEG (proc. Number 201810267000023). Instituto Tecnológico Vale (ITV) supported this publication. The funders had no role in study design, data collection and analysis, decision to publish, or preparation of the manuscript.

## Grant Disclosures

The following grant information was disclosed by the authors:
Conselho Nacional de Desenvolvimento Científico e Tecnológico (CNPq): 142457/2014-0.
Programa Nacional de Cooperação Acadêmica (PROCAD 2013): 071/2013.
Conselho Nacional de Desenvolvimento Científico e Tecnológico (CNPq): 304494/2019-4 and 302611/2018-5.
MCTIC/CNPq: 465610/2014-5.
FAPEG: 201810267000023.
Instituto Tecnológico Vale (ITV).

## Competing Interests

Daniel de Paiva Silva is an Academic Editor for PeerJ.

## Author Contributions

- Cássia de Carvalho Teixeira conceived and designed the experiments, performed the experiments, analyzed the data, prepared figures and/or tables, authored or reviewed drafts of the paper, and approved the final draft.
- Leonardo Carreira Trevelin conceived and designed the experiments, analyzed the data, authored or reviewed drafts of the paper, and approved the final draft.
- Maria Cristina dos Santos-Costa conceived and designed the experiments, authored or reviewed drafts of the paper, and approved the final draft.
- Ana Prudente conceived and designed the experiments, authored or reviewed drafts of the paper, and approved the final draft.
- Daniel Paiva Silva conceived and designed the experiments, analyzed the data, authored or reviewed drafts of the paper, and approved the final draft.

## Data Availability

  The dataset used (occurrences) in our analyses relied on the published data of Ribeiro-Júnior & Amaral, 2016, available at: https://www.tandfonline.com/doi/full/10.1080/23766808.2016.1236769.
  The code used in our modelling procedures is available at GitHub: *Andrade, Velazco & De Marco Júnior, 2020*, code repository https://github.com/andrefaa/ENMTML.

## Supplemental Information

Supplemental information for this article can be found online at http://dx.doi.org/10.7717/peerj.13028#supplemental-information.

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
