# Peer review of "Synergistic effects of climate and landscape change on the conservation of Amazonian lizards"

_PeerJ, doi:10.7717/peerj.13028_

## Round 0.1 · original submission · Major Revisions

Thank you for your submission to PeerJ. Three expert reviewers have provided detailed and thoughtful comments on your manuscript. The aims of this manuscript and study are important and offer some interesting and exciting outcomes. However, the reviewers highlight that there are some potential inconsistencies with the statistical analyses chosen, which should be addressed and justified upon resubmission. Please address the reviewers comments carefully, directly, and completely. I look forward to receiving a revised manuscript.

Reviewer 1 ·

Basic reporting

The article fails to meet standards for one (the first) of the four criteria listed for Basic Reporting. Mainly, but not exclusively, the introduction and methods (especially), were challenging to understand and needed much rewording, more specific language (general or abstract language was common throughout the manuscript), and addition and reorganization of sentences/paragraphs to improve clarity and logical flow of information. Generally, the text information appears well supported with citations except where noted (very infrequent). Generally, article structure was good; appropriate figures and tables, including raw (or at least supporting Supplemental data sets derived from raw data) supporting objectives were provided although data in Tables S3 and S4 could be summarized and provided in other main text tables to aid interpretation of the results by the reader. Modifications to Figures 1 and 2 are suggested. I particularly like Figure 3. Please see specific comments for each section of the manuscript provided below and inserted within the pdf manuscript document itself. By appropriately addressing all of the attached outstanding comments, I think the article should then be considered for a "pass" by PeerJ.

INTRODUCTION
The introduction would benefit from some moderate reorganizing and added details (about climate change effects on Amazon lizards). Some sentences required further clarification or rewording to be clearly understood.

I suggest rearranging text about SDMs of lines 74-79 by moving it nearer to the bottom of the Introduction, where you begin to discuss your research objectives on line 112. If you can combine the SDM paragraph with the one beginning on line 106 into 1 paragraph that should improve the logical flow of your Introduction.

Either at line 80 or line 99- Insert a new paragraph ahead of the one that is there. The new paragraph should describe the literature about climate change in Amazonia region and the resulting effects of those changes on range shifts or other projected impacts to lizards in that region. A minor suggestion- wherever you decide to insert the paragraph, I suggest you re-arrange the order of listing “climate change” before or after habitat loss/fragmentation wording when referencing them together, to match the order they are presented in paragraphs (for example, in line 63).

MATERIALS & METHODS SECTION

Sentences should be reorganized within and among paragraphs in this section and including the addition of a new subheading for a climate projections section and/or a scenarios section that fully describes the climate projections and levels of deforestation in each scenario.

Paragraph starting on line 130- the paragraph is missing a strong topic sentence summarizing the paragraph as a whole; please add one. You may need to add other following sentences to lead into the current one on line 130. Additionally, some sentences should be reorganized to reflect sequence methods were conducted (see specific comments in pdf).

Lines 137-139- important- What is meant by "related to"? Can you reword so that the relationship isn't about the centroid but with the muni itself? Reword to something like “occurrences were located within municipalities”. What exactly does it mean to be related to a muni ecologically? Because I thought a muni reference in the sentence above was only used to determine lizard locations and didn't have any real ecological significance but this sentence seems to make a distinction b/w being in a muni or not but it's not clear why it matters. It appears you forgot to add some sentences with details about the significance of being in a muni vs. not.

RESULTS SECTION

Line 253-254- It’s unclear what “V” is. Is it year? If so, 2016 isn’t the future (future identified as 2050 in methods) and 1953 isn’t current. P-values should be accompanied by test statistics values. Most importantly what is the effect size (with units)? Table S3 (cited line 252) only provides unsummarized results for these reported values.

Lines 264-5- It would help the reader for you to label these cited locations on Fig. 2 maps.
Line 269- citation Table S4- as with table S3, a table would be helpful that summarizes the data (distills it down), perhaps by grouping species in a way that highlights main points for conservation that you present in abstract and main text and by calculating the these percents for each group.

Line 297-299- I suggest providing the effect size (change in area between present and scenarios) and test statistic info that accompany the P-values in Table S5. I also suggest reporting in the text, the mean effect sizes for averages across all the 4% (mean across all these) and 19% (mean across all these) of species respectively.

Fig. 1- For panel (C) optimistic and pessimistic scenarios have very similar curves. If this panel only serves as an example and not actual results of the modeling, I suggest you adjust the curve for one or the other scenario so that optimistic is more “optimistic” and pessimistic is more “pessimistic”, or use a lizard species with more divergence in scenarios as an example, to better distinguish the 2 scenarios.

DISCUSSION SECTION

All comments are inserted in manuscript pdf. Some sentences in text require more detail/specifics and suggested rewording for sufficient clarity about authors’ intended meaning.

CONCLUSIONS SECTION

See comments inserted in manuscript pdf- minor text revisions and suggested rewording.

Experimental design

The article failed to meet standards for the 4th criteria because key information about the SDM, climate projection, and deforestation modeling processes were lacking sufficient detail as described more below. It's uncertain whether the article fails the 3rd of the 4 criteria because methods were inadequately described in enough detail to determine whether scientific rigor was sufficiently achieved. By appropriately addressing all of the attached outstanding comments, I think the article should then be considered for a "pass" by PeerJ.

MATERIALS & METHODS SECTION


Lines 134-137- Can the authors explain how well the habitat attributes of distributions of the lizards that were used in SDMs (training and validation data sets) represent the habitats of those lizards when they were collected? The concern is that if habitats have changed greatly since collections, that habitat determined using recent mapping technology (it’s still unclear to me how baseline/current attributes of habitat were determined- PRODES satellite imagery on line 177?) may not be the habitat that was suitable for those lizards during collections, especially if lizards are in areas/municipalities where lots of deforestation and development has been occurring for decades. I think this is a very important issue the authors need to address because SDM results could misrepresent areas of relatively high deforestation as being suitable areas for lizards, when at the time of collection there was a lot more forest.

Lines 141-144- I'm not sure I understand. Please reword. Did you interpolate your predictor variables from bioclimatic variables in WorldClim? You should define variables from WorldClim that you used to create your predictor variables, define the variables that were created by them (i.e., your predictor variables), and describe in enough detail how those predictors were created and used in your analysis. All those things don't necessarily need to be included at this point in your paragraph if some of it would logically flow better somewhere else, say in the analysis for example.

Lines 144-150- you describe a little, only one climate scenario here, yet on lines 230-231 you indicate 3 scenarios. These lines are confusing because you refer to using “all” pessimistic projections of RCP8.5 from apparently multiple climate models (which ones specifically?) and it’s unclear how you combine these projections for your scenarios (are these ensemble averages?) and what variables you use (temp, precip)?

I think you need to add a new section and subheading to describe the climate projections used for your modeling scenarios in your analysis and justify why you used these and include which climate scenarios, variables, how they were extracted for each species occurrences or average latitude/longitude locations for each species, and any other details the reader would need to understand how you created your modeling scenarios.


Lines 154-158- More detail here is needed. Why were all 3 models needed? And how are “RBF” and “C = 1” important in the modeling and why these values were used and not others? Also, what are the predictor variables and response variables used in these models- these need to be fully described here and using table(s).
Line 159-160- Why would you remove all highly correlated variables? I think you’re not describing a step- selection among those correlated variables to include them in your models (because you don’t need to remove all of them only some of them that are correlated with the others). This comment applies to all Supplemental tables and figures of Table S1 is the first: there are no table headings or figure captions to describe what, how, when, where of each, making it hard to understand what they apply to (not stand-alone either); I’m not sure what PeerJ criteria are but this seems like a deficiency.

Lines 166-168- Reword and clarify. How did you average ranges across scenarios? Which scenarios? If you introduced the scenarios in a new section above then you won’t need to define them unless it was a subset of scenarios you averaged. Define “consensual map”; how was there consensus? by whom?

Paragraph for lines 175-191- This paragraph needs more clarification in multiple places. Has this deforestation model been published? If so, provide citations in the first sentence. If not, has it undergone stakeholder review or other review for accuracy, realism, performance, etc.? Would the Soares-Filho et al. 2006 citation apply toward this model so that you could cite it in the first sentence? If there hasn’t been any technical review other than from those that developed the model, this is concerning for your analysis. What is the PRODES imagery, define it? To make your BAU and GOV scenario deforestation rates comparable, it would help if you could add a little extra text translating the 55% reduction in deforestation to the “expected forest reduction” as you cited for the BAU scenario. Simply, would that be calculated as 0.37 * 0.45, or BAU rate * rate relative to the BAU rate? I’m having a very hard time understanding the last 2 sentences (lines 186-191) to be able to provide helpful comments for these lines. If you are extrapolating a models results to an area outside of its domain area, it’s important for the reader to know how you did that and understand the limitations with the new resulting area it’s extended to.

Paragraphs for lines 207-218- If the criteria of Rodrigues et al. 2004 was indeed “adapted” and not “adopted” then you should describe what changes were made for the criteria. As written (lines 209-210), it’s unclear what the criteria is measuring. Why species with <1000 km2 of stable habitat area (this could range down to 0 km2 by the way; I think you need a lower limit to put a cap on it) would have 100% of their distribution in PAs, >250 with 10% in PAs (again this could range to >1000 so not sure how these levels are mutually exclusive). In other words, I’m not sure the logic for these percentages by areas of stable habitat area. There should be some explanation of the criteria indicated I think. Lines 214-18: “all categories of PAs”; I don’t recall these being defined above and you probably need to do it here if not above.

Lines 194-218 or 221-239- somewhere in one of these sections you should briefly describe the process for identifying patterns 1-5 relating to Figure 3. I know you go into some general detail in describing the patterns in the results section but you didn’t provide background info or explain criteria/observations on how you determined these patterns that should probably reside in the methods. Similarly, you didn’t describe the process in using future scenarios to map these patterns in panels C and D. For example, it is unclear whether C and D are the product of the pessimistic and optimistic scenario results being combined in some manner to produce a single pattern 3 and single pattern 5, or did you map 3 and 5 only for the pessimistic scenario because this is the only future scenario presented in figure 3 (panel B). In other words, I would expect 2 sets of pattern results- 1 each for pessimistic and optimistic, but there is only 1 set and it’s unclear which scenario the patterns belong to or if it’s some joint combination (average)? How you created the pattern results in figures 2 and 3 need to be clearly described in the methods.

Lines 230-31- “Thus, we produced three curves for each species: one for the present day (2019) and two for the 231 future (2050), considering pessimistic and optimistic scenarios.”
This reminds me that you need to create a paragraph that describes each scenario thoroughly and perhaps a table that lists each current period, future optimistic, and future pessimistic scenario and their attributes, including the type of climate projection and deforestation scenario that characterizes each one (a full description of each).

Validity of the findings

Assuming that the study was conducted with suitable and well-designed methods (high rigor, 3rd criteria of 2. Experimental Design above), the article would appear to meet all 4 validity of findings criteria. Below are specific comments as well as comments provided in the attached pdf document. By appropriately addressing all of the attached outstanding comments, I think the article should then be considered for a "pass" by PeerJ.

RESULTS SECTION

Lines 260-261- You report results for pessimistic scenario but not other 2; report them as well.

Fig. 2- Why are there no results for the optimistic scenario? Please add optimistic scenario results, in which case you will have a new panel (in total panels A-E). Please see comments above for the methods section, lines 194-218 or 221-239. Some brief text from the methods that you use to address those comments will need to be inserted in these figure captions to describe which scenario(s) are presented in these figures.

CONCLUSIONS SECTION

Lines 420- Circling back to an earlier comment, habitats modeled in SDMs may underrepresent the natural forest land cover needed to support lizards because of the temporal lag between when lizards were collected for scientific collections and when the landscape was characterized for this study. Unless I’m mistaken (which is possible but methods were not detailed enough for me to determine the timing of mapping), you should describe this problem with the analysis and how that could also affect the results to then affect conservation of lizards as you did in the previous lines.

Annotated reviews are not available for download in order to protect the identity of reviewers who chose to remain anonymous.

Reviewer 2 ·

Basic reporting

Suitable.

Experimental design

Suitable, except for some issues on Methods, as described on the fourth section.

Validity of the findings

Suitable, except for some issues on statistical procedures, as described on the fourth section.

Additional comments

The manuscript “Synergistic effects of climate and landscape change on the conservation of Amazonian lizards” provides a pertinent and important approach to how climate change and landscape changes could affect 63 species of Amazonian lizards. The introduction and hypotheses of the work are very clear and well grounded! Material and Methods is well-written, but there are points I would like to see revised and better clarified: (i) I think that updates on climate variables and future climate scenarios from the WorldClim database (CMIP6 rather than CMIP5) should be incorporated into this study; (ii) considering that there is a widespread recommendation that the construction of potential species distribution models be carried out based on at least 25 occurrence records (or, alternatively, that models with less than 25 records be evaluated in a peculiar way, such as jackknife “leave-one-out”), how do the authors justify the use of at least 10 records?; (iii) the authors used 3 Machine Learning algorithms, but they do not justify not using other classes of algorithms. Why did they only use this class of algorithms?; (iv) lines 144-150: despite saying that they used only the pessimistic scenario (RCP8.5), results are presented for both the optimistic and the pessimistic scenarios - please revise; (v) considering that the present study has a conservationist implication, what is the advantage of using the Receiver—Operator Curve” (ROC) as a threshold?; (vi) the AUC metric is out of use as a way of evaluating models in SDM, due to its lack of robustness. I suggest using TSS or another (in the case of models with less than 25 records, jackknife could be an alternative). In conclusion, line 419, please state which other environmental factors were not considered and which would also be important for the models.

Reviewer 3 ·

Basic reporting

Dear editor,

I am very thankful to review the article entitled “Synergistic effects of climate and landscape change on the conservation of Amazonian lizards”. The article talks about the synergistic effects of landscape modification, climate change on lizards species in the Amazon. The article is well written and highlight the urge needs to amazonian conservation. But I have some doubts and question that I will be listing below.

Experimental design

The processes modeled by author occurs in the different temporal and spatial scales. Climate change will be occur on a broad-scale perspectives (spatial) and around 50 years (time). The landscape changes act on biodiversity on a narrow-scale (spatial) and in few year (time). These difference in processes acting on biodiversity cause difficult to understanding the synergistic effects. In my opinion, the authors do not use spatial and time in the correct form: Normally, climate and landscape are used as ecological filtering, here I don't see it.

I) I) the temporal climate models and landscape variables in not the same. You inferred the effects of climate change on species richness and after use landscape variables, but in the future this landscape variables is not real. It is a simplistic scenario that can be changed depending of a policy decision, causing noise on models results and conservation discussion. Why did you used models only under current landscape and climate scenario using a hierarchical approach (climate broad and landscape narrow), is sound good and the results is important to discuss amazonian conservation. I also believe that more landscape metrics is important your models building.

II) the species richness is related not only habitat amount (habitat area). The species (the basis of species richness) suffer with landscape configuration as connectivity, heterogeneity, matrix quality and others. So, use only a single landscape variable (habitat area) is not a real scenario to effect of landscape modification on species richness. I suggest the use o other landscape variables to simulate the landscape changes effecting on species richness. Only habitat area is unreal and can be causing extintic debt noise - more predict species than the real (see species x area relation and species debt).

Iii) Why did you use of Pearson correlation? Why 0.8 as threshold? I suggest the use of PCA eigenvalues as variables. See De Marco & Nobrega 2011 and the ENMGadgets R package.

iv) why did you use only Maxent and RF? You used only machine learning methods? Why? The machine learning sounds good to model species with broad distribution. But species with restricted distribution others algorithms are better.

Validity of the findings

Based on my comments, I would like to see a novel analysis approach and novel findings.

---

## Round 0.2 · Minor Revisions

Thank you for your resubmission - your revised manuscript has been reviewed by two experts in the field, and I implore you to take their useful comments into account for your next submission. Please make the appropriate text edits to the manuscript, and I look forward to your resubmission.

Reviewer 1 ·

Basic reporting

I suggest that a "Pass" for basic reporting be assigned to this manuscript. The authors addressed all of my most substantive comments of my initial review. There are only a few relatively minor comments I suggest that the authors address. I provided one of the main comments below that follows on the authors' rebuttal response (In order you will see my comment from my first review, authors' response, then my follow-up response to theirs. Other minor comments and suggested edits can be found in the attached manuscript PDF with my comments included.

Reviewer 1, first review: "Lines 137-139- important- What is meant by "related to"? Can you reword so that the relationship isn't about the centroid but with the muni itself? Reword to something like “occurrences were located within municipalities”. What exactly does it mean to be related to a muni ecologically? Because I thought a muni reference in the sentence above was only used to determine lizard locations and didn't have any real ecological significance but this sentence seems to make a distinction b/w being in a muni or not but it's not clear why it matters. It appears you forgot to add some sentences with details about the significance of being in a muni vs. not."

Authors' response: "We agree with the reviewer that the wording in these lines was confusing. We meant to talk about the resolution of some geographic coordinates used as occurrence records. Therefore, we re-wrote these sentences and re-allocated them in the previously mentioned new subheading, “Species occurrence data”. "

Reviewer 1 follow-up review: "Previous review comments about municipalities and centroids (Why species locations are referenced based on coarse municipality areas?, what it means for occurrences to correspond?) were not fully addressed here either. Please address.

I found some partially clarifying information, only by reviewing one of your cited references. Below is what I found in Ribeiro-Júnior, M. A., & Amaral, S. (2016a). You may need to summarize this information for yours? Or otherwise add pertinent details about specimen locations.

"Geographical coordinates associated with the examined specimens were used when available, otherwise specimen locations were georeferenced based on collector’s field notes, personal information, or any other source that could identify the locality with confidence. In such cases, coordinates were identified with gazetteers (e.g. [22]) and Google Earth [34]. After searching every available gazetteer to identify the precise geographical position of a record, when specific localities could not be retrieved, the record was not considered for analysis. Records with incomplete information (e.g. Pará, Brazil) were still cited in Appendix 1 and used for distributional purposes."
"Ribeiro-Júnior, M. A., & Amaral, S. (2016a). Diversity, distribution, and conservation of lizards
699 (Reptilia: Squamata) in the Brazilian Amazonia. Neotrop. Biodivers. 2, 195–421.

"

Experimental design

I suggest that a "Pass" for experimental design be assigned to this manuscript. The authors addressed all of my most substantive comments of my initial review. There are only a few relatively minor comments I suggest that the authors address. I provided one of the main comments below that follows on the authors' rebuttal response (In order you will see my comment from my first review, authors' response, then my follow-up response to theirs. Other minor comments and suggested edits can be found in the attached manuscript PDF with my comments included.


Reviewer 1, first review: "Paragraphs for lines 207-218- If the criteria of Rodrigues et al. 2004 was indeed “adapted” and not “adopted” then you should describe what changes were made for the criteria. As written (lines 209-210), it’s unclear what the criteria is measuring. Why species with <1000 km2 of stable habitat area (this could range down to 0 km2 by the way; I think you need a lower limit to put a cap on it) would have 100% of their distribution in PAs, >250 with 10% in PAs (again this could range to >1000 so not sure how these levels are mutually exclusive). In other words, I’m not sure the logic for these percentages by areas of stable habitat area. There should be some explanation of the criteria indicated I think. Lines 214-18: “all categories of PAs”; I don’t recall these being defined above and you probably need to do it here if not above."

Authors' response: "We changed the wording to address the suggested points. The exception is that the criteria are cited exactly as Rodrigues et al. (2004) proposed, so we maintained it. "

Reviewer 1 follow-up review: "There was an error in “250” (revise). See comment on PDF document line 264- comment: “Revise, this should be "250,000" (replacing decimal with comma), which I discovered after looking up Rodriguez et al. (2004). After revising, this set of criteria will now makes sense. Initially, I thought 250.000 was just an overstatement in the value's precision.”

Otherwise, author revisions have addressed these comments.
"

Validity of the findings

I suggest that a "Pass" for validity of the findings be assigned to this manuscript. The authors addressed all comments of my initial review.

Additional comments

Dear Authors,
I enjoyed reading your revised manuscript. Thank you for your effort in addressing my initial comments and I provided only a few minor comments for additional clarification of your methods plus a few minor edits, which you can find in the attached manuscript PDF with my comments. Please make the appropriate text edits in response to my comments, though I see NO reason to review your next manuscript draft.
Best regards,
Reviewer 1

Annotated reviews are not available for download in order to protect the identity of reviewers who chose to remain anonymous.

Reviewer 2 ·

Basic reporting

No comment.

Experimental design

I thank the authors of the manuscript “Synergistic effects of climate and landscape change on the conservation of Amazonian lizards” for the corrections and justifications for my comments. I would like to ask you, please, to check the possibility of inserting in the text the justification given in the reply letter for the use of the Receiver Operator Curve (ROC) as a threshold. Added to that, considering the current advantages of remote access to workstations, I insist on the possibility of redoing climate models according to CMIP6, possibly even including them as a comparison with CMIP5.

Validity of the findings

No comment.

Additional comments

No comment.

---

## Round 0.3 · Minor Revisions

Dear authors,

I appreciate the time you spent editing this manuscript for resubmission. However, I do not believe that because something is too much work or you are overburdened is an adequate excuse to justify not performing additional analyses. Sometimes analyses are incorrect and should not be published if they are not going to be redone. There needs to be a logical explanation as to the validity of your current analytical methods. They need to be sound methods, and here in this manuscript, that means your analytical methods will need to be justified more specifically than they currently are.

You have begun approaching this explanation in your rebuttal letter but there is no direct address of it in the manuscript itself, and there needs to be. Perhaps this could be placed in a section about study limitations in the discussion.

Reviewer 1 was always more supportive of your work than the other two reviewers - but that does not mean that Reviewer 1's comments and opinions are more valid than the other Reviewers. 2 and 3 (who did not provide a second review) present valid criticisms that should be directly presented in the manuscript itself. Both take some issue with the analysis performed. I'm not saying for certain that your analysis has been incorrectly performed, but I am saying that it needs to be defended and clarified directly in the manuscript itself. It might be worthwhile adjusting the manuscript directly with your responses to reviewer 3's comments. I looked back through your submission and rebuttal letters and it appears that you have only addressed these points in the rebuttal letter, but not in the manuscript. While I had mentioned this ever so briefly in my last decision letter [quoted as "Please make the appropriate text edits to the manuscript"], I think it is worthwhile going through the previous revision comments and addressing them in the MS itself.

A theme with your response to R3 in version 1 was that their suggestions would make excellent new studies/future directions. It is worth spending more time on that idea in the manuscript itself... overviewing the limitations and suggesting future project ideas. Sometimes this is necessary for prediction modelling because data is consistently updating and modelling practices are consistently improving. If you had done that, I suspect R2 in this version would have been more forgiving with their comments. It is well within your rights as an author to reject a reviewer's suggestion - which is what I was expecting, as I had suggested minor revisions for this version of the manuscript and redoing analysis would justify a Major revision or a manuscript rejection. However, changes to the manuscript itself NEED to be included such that the criticism is no longer valid.

I am suggesting minor revisions, again. Please address these points, and perhaps go back through R2 and R3's past comments on version 0 to help you direct you. Both reviewers requested additional analyses; therefore, it's likely that the general readership might have these same questions.


As an additional note for handling manuscript revisions. It helps the editor substantially understand where the manuscript has been edited and adjusted, and to determine if the edits are sufficient if the authors include line numbers and direct quotes from the manuscript in the rebuttal. Please provide line numbers and quote any changes you complete that are relevant to these comments directly in the rebuttal letter.

---

## Round 0.4 · accepted · Accept

Thank you for adding the additional points in the discussion. The manuscript is now suitable for publication, congratulations!